# Ability and accuracy of the smartphone-based O`VIEW-M® sperm test: Useful tool in the era of Covid-19

Kyu Shik Kim[1]☯, Ji Hoon Kim[2]☯, Ji Hyoung Roh[3], Daegwan Kim[3], Hyang Mi Kim[4], Jung Ki Jo[1,2]*

1 Department of Urology, Hanyang University College of Medicine, Seoul, Korea, 2 Department of Medical and Digital Engineering, Hanyang University, Seoul, Korea, 3 Medical Device Development Center, Daegu-Gyeonbuk Medical Innovation Foundation, Daegu, Korea, 4 LG Sciencepark AI, Seoul, Korea

☯ These authors contributed equally to this work.
* victorjo38@hanyang.ac.kr

**Data Availability Statement:** All relevant data are within the paper and its Supporting Information files.

## Abstract

Male infertility affects up to 12% of men. Although manual testing using microscope examination and computer-assisted semen analysis are standard methods of measuring sperm count and motility, these methods are limited by being laboratory based. To investigate the usefulness of a novel semen analysis device using a smartphone camera. This prospective multicenter randomized parallel design trial enrolled 200 men aged ≥19 years of age between August and December 2018. Each subject was advised to use the Smart Sperm Test for OVIEW-M at home after 5 days of abstinence. The accuracy of the OVIEW-M test relative to the in-hospital test was determined. A questionnaire was administered to assess subject likelihood of using the OVIEW-M. Measurements using standard methods and the OVIEW-M showed similar sperm counts and similar motile sperm counts. Correlation analysis showed significant correlations between sperm count and sperm motility when measured by OVIEW-M tests (r = 0.893, p < 0.01) and standard microscope examination (r = 0.883, p < 0.01). Of the subjects who responded to questionnaires, 43% regarded the results of the OVIEW-M tests as reliable and 18% as unreliable. Semen analysis with the smartphone-based application and accessories yielded results not inferior to those of laboratory tests. Men who visit the hospital for evaluation of infertility can easily perform OVIEW-M semen tests at home.

## Introduction

Male infertility is a serious global problem, with around 485 million couples experiencing infertility [1]. The causes of infertility have been identified in 15% of couples, with the incidence found to be equal in men and women [2]. Although 2.5% to 12% of men worldwide are thought to be infertile, estimates of male infertility in many countries are inaccurate. Most surveys are of couples and of women who are trying to have children. In contrast, causes of infertility cannot be easily determined in countries that allow male supremacy or polygamy due to

**Funding:** This research was supported by the Bio & Medical Technology Development Program of the National Research Foundation (NRF) & funded by the Korean government(NRF-2019M3E5D1A01066057). Prof. Jung Ki Jo. The funders had no role in study design, data collection and analysis, decision to publish, or preparation of the manuscript.

**Competing interests:** JHK is the Chief Executive Officer of INTIN Corporation. The products in development or marketed products associated with this research are as follows. A measuring device of salive for smart phone (Reg. No. 10-1533343), the test device for Body fluid analysis using natural light (Reg. No. 10-1813866), CHAMBER FOR INSPECTING BODY-FLUID AND BODY-FLUID INSPECTING APPARATUS USING THE SAME (Reg. No. 10-0029588). The O'VIEW-M® Sperm Test is developed by the INTIN Corporation. This does not alter our adherence to PLOS ONE policies on sharing data and materials. All authors declare that there is no further competing interests.

cultural differences. Historically, male infertility was not recognized as a disease until recently, and definitions of male infertility differ among countries [1].

Semen test criteria for male infertility based on the World Health Organization (WHO) classification [3]. Semen tests measure sperm count, motility, and malformation [4, 5]. Causes of male infertility have been classified into those involving the hypothalamus or pituitary gland, the testicles, or sperm delivery by sex glands associated with the penis and genitalia [6].

In the Republic of Korea, the declining fertility rate is becoming a social problem. Sperm counts may be reduced or sperm damaged due to various causes, such as environmental pollution and the use of electronic devices during the process of industrialization. Causes of male infertility include oligozoospermia (low-spermatozoa count), asthenozoospermia (poor sperm motility), and teratozoospermia (abnormal sperm morphology). Abnormal semen measurements are idiopathic in 30–40% of infertile men. Although treatment of some men may enhance the likelihood of pregnancy, other men are resistant to treatment [7].

Screening for male infertility requires measurements of sperm counts and sperm motility. Men may be uncomfortable visiting a professional medical institution for diagnosis of male infertility. The OVIEW-M is a self-testing device that can measure the number and activity of sperm using an optical camera on the front side of an individual's smartphone. The accuracy of the OVIEW-M relative to standard laboratory methods for measuring sperm count and motility has not been determined. This clinical therefore compared the accuracy of the OVIEW-M with that of standard methods.

## Materials and methods

### Study design

This prospective multicenter randomized parallel design trial enrolled 200 men aged >19 years of age from August to December 2018. All subjects provided written informed consent using a form developed by one of the investigators. Subjects were administered a questionnaire that evaluated ejaculation and use of contraceptive devices within 5 days; history of vasectomy and other contraceptive procedures; medical history within the previous 6 months, including surgery; and medications being taken.

Subjects were included if they were healthy male adults aged ≥19 years, had no specific disease that could affect sperm count or motility, were not taking hormone medications, had not been assessed during the previous 5 days, and provided written informed consent. Females were excluded, as were males aged <19 years and those unable to provided written informed consent. Also excluded were men with congenital or genetic diseases affecting spermatogenesis and nocturnal emission, and subjects who had participated in other clinical trials within the previous 30 days.

While in the hospital, subjects were asked to provide a >1 cc semen sample by masturbation in an uncontaminated cup. Subjects were instructed to use the Smart Self-Sperm Test (OVIEW-M) at home after 5 days of abstinence at home. The accuracy of the OVIEW-M test was assessed by comparison with standard in-hospital methods. After the test, a questionnaire was administered to each subject to determine their likelihood of using the OVIEW-M.

### Overview of the OVIEW-M device and testing procedure

The OVIEW-M is a self-testing device that measures the number and activity of sperm using an optical camera on the front side of an individual's smartphone. It consists of a collecting cup and stick and a testing device composed of a cover, chamber, and body, which is connected to the smartphone.

Each test was performed using both an Android smartphone and an iPhone, with the OVIEW-M application downloaded from Google Play and the iPhone App Store, respectively. While at the hospital, each subject tested the device and application before returning home to confirm that the application was working properly and that the testing device was visible to the camera.

### Standard semen testing procedure

A semen sample in a disposable cup was allowed to separate for 15 minutes, as above, and a drop placed at the center of a 1 cc slide glass using a pipette. The sample was covered with a cover slip and lightly moved to spread the sample evenly, and the slide was viewed on an optical microscope (400×). The numbers of spermatozoa and of active spermatozoa were determined using a grid.

### Ethics statement

This study was approved by the clinical study ethics committee (IRB) of the institution performing the clinical trial (MPUC-A-1808-S-01). The trial was conducted in accordance with the standards of the KGCP (Clinical Trial Management of Medical Devices) and the principles of the Declaration of Helsinki.

### Statistical analysis

To determine whether the use of standard methods by specialists at different medical institutions yielded similar results, semen samples were randomly sampled and assayed by laboratories at 13 regional medical institutions in the Manpower Urology Network, a representative urology clinic in Korea. To verify the difference between standard and OVIEW-M methods, 210 people were recruited, based on a 5% drop out rate. However, because no subjects were excluded or dropped out, 200 subjects were recruited. Although most hospitals evaluated more than 15 subjects, some did not meet this quota, with data from these subjects being replaced by those from hospitals that evaluated more than 15 subjects used. Results obtained using the OVIEW-M device and standard method were compared by Student's t-tests. All statistical analyses were performed using SPSS for windows, version 20.0.

## Results

### Design of optical equipment using low power

Briefly, a semen sample was added to the sampling cup and mixed well with a stick. The sperm and suspended matter were allowed to separate for 15 minutes to reduce the viscosity of the semen. The semen was placed in the square section of the chamber of the testing device, and the chamber was placed between the cover and the body of the device. The camera was connected and the OVIEW-M application started on the smartphone (Fig 1). While the camera was running, the light source was centered using the external stand on the camera screen. The testing device was positioned on the front camera, and the measurement start button was pressed. Results were displayed within 10 seconds.

### Validity evaluation (accuracy evaluation)

Mean sperm counts ($49.49 \pm 31.10 \times 10^3$/μl vs. $49.6 \pm 31.06 \times 10^3$/μl; t = 0.68, p = 0.50) and sperm motility ($49.59 \pm 24.72 \times 10^3$/μl vs. $49.45 \pm 24.44 \times 10^3$/μl; t = .66, p = 0.51) did not differ significantly when measured by OVIEW-M tests and standard microscope examination (Table 1). Correlation analysis showed significant correlations between sperm counts and

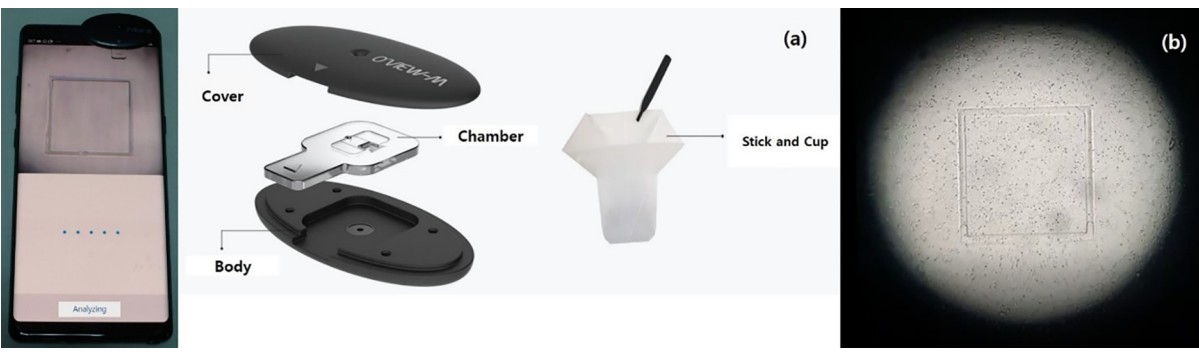

**Fig 1. Installed OviewM and View of Sperm analysis using Oview M.** (a) Consist of Oview M (b) View of Sperm analysis using Oview M.

sperm mobility when measured by OVIEW-M tests (r = 0.893, p < 0.01) and standard microscope examination (r = 0.883, p < 0.01). These results confirm the reliability of the OVIEW-M measurement tool in assessing sperm count and sperm motility.

## Questionnaire analysis

Of all respondents, 34% planned to use the smart self-sperm OVIEW-M test to measure sperm count and sperm motility in the future, with 76% of the respondents being optimistic about the results obtained with this device (Fig 2). Although 43% of respondents regarded the results obtained with OVIEW-M test as reliable and 18% did not, most respondents did not have sufficient confidence in the results of the smartphone test (Fig 3).

## Discussion

This study showed that an application using a smartphone and accessory parts could accurately determine sperm counts and sperm motility in semen samples, when compared with results obtained by cell counting in hospital laboratories test. These results were reliable and may be used to assess confirm sperm counts and sperm motility not in infertile men, but also in men who have undergone vasectomy and vasovasostomy.

Smartphones were reported to be used by up to 96.8% of subjects in developed countries in 2015 [8]. Smartphones have powerful computing systems and internet access, making them useful in many applications, including the facility to attach low-cost microscopic devices [9].

The development of micro-technology and the rapid growth of consumer electronics have laid a solid foundation for the development of mobile health technology that can change the paradigm of world health today. In daily life, we can exercise with using smartphone with measuring vital signs, total calories that we consumed, and distance of we jogged. Now, many research showed application of smartphone in medical field with various kind of ways. We can

**Table 1. Repeatability, precision, and correlation coefficients of motile sperm concentration and motility.**

| | | N | Mean ± SD, x $10^3$/μl | Mean ± SD, difference, x $10^3$/μl | *p*-value | Correlation coefficient (concordance) | *p*-value |
|---|---|---|---|---|---|---|---|
| Sperm Count | | | | | | | |
| | Microscope analysis | 200 | 49.60±31.08 | 0.11±2.29 | 0.50 | 0.997 | <0.01 |
| | OVIEW-M device | 200 | 49.49±31.10 | | | | |
| Sperm Motility | | | | | | | |
| | Microscope analysis | 200 | 49.45±24.43 | -0.14±2.91 | 0.51 | 0.993 | <0.01 |
| | OVIEW-M device | 200 | 49.59±1.75 | | | | |

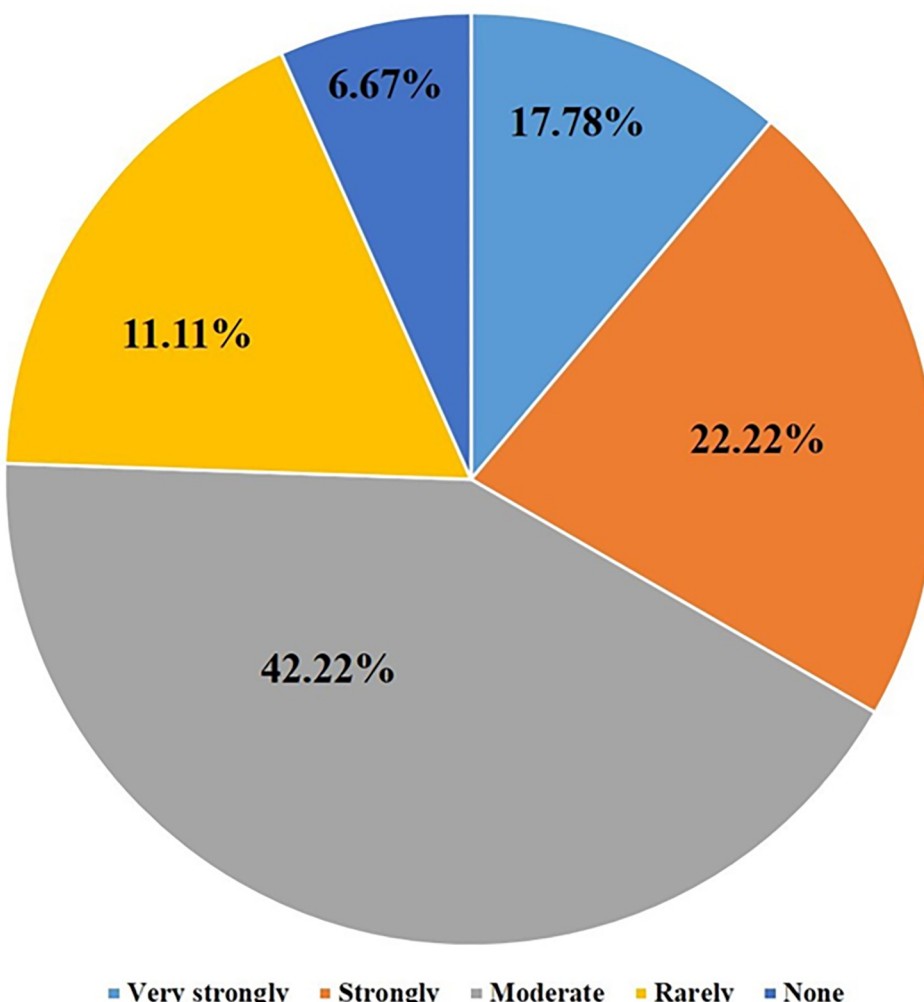

**Fig 2. Desire to buy O'view M (smart phone based self- semen analysis tester).**

use smartphone application using fractal image analysis of moles [10]. Also, Add on lenses on smartphone is feasible for checking up macrovesicular steatosis in liver allograft biopsies [11]. Smartphone applications in urology include the University of Washington Peyronie's Examination Network smartphone (UWPEN), which was developed to assess penile angulation in a home environment [12]. Other smartphone applications include a voiding diary, the tracking of "forgotten" ureteral stents, and applications to manage enuresis and ureteral stones [13–15]. Smartphone applications for diagnostic purposes include a guide for surgical treatment using three-dimensional and virtual reality technology [16]. In addition, smartphone applications used in combination with various technologies combined with are being developed to assist diagnosis and treatment [17, 18].

To date, almost all semen samples have been tested in clinical laboratories by a specialist. Standard tests include measurements of the number of spermatozoa, their mobility and their deformity. The results of these tests have been used to diagnose causes of male infertility and to analyze the effects of treatment. Couples who wish to get pregnant before going to a hospital can search the medical literature and the internet to find a pregnancy tester that assesses the causes of infertility. To date, however, no tool or method outside a hospital is available to test

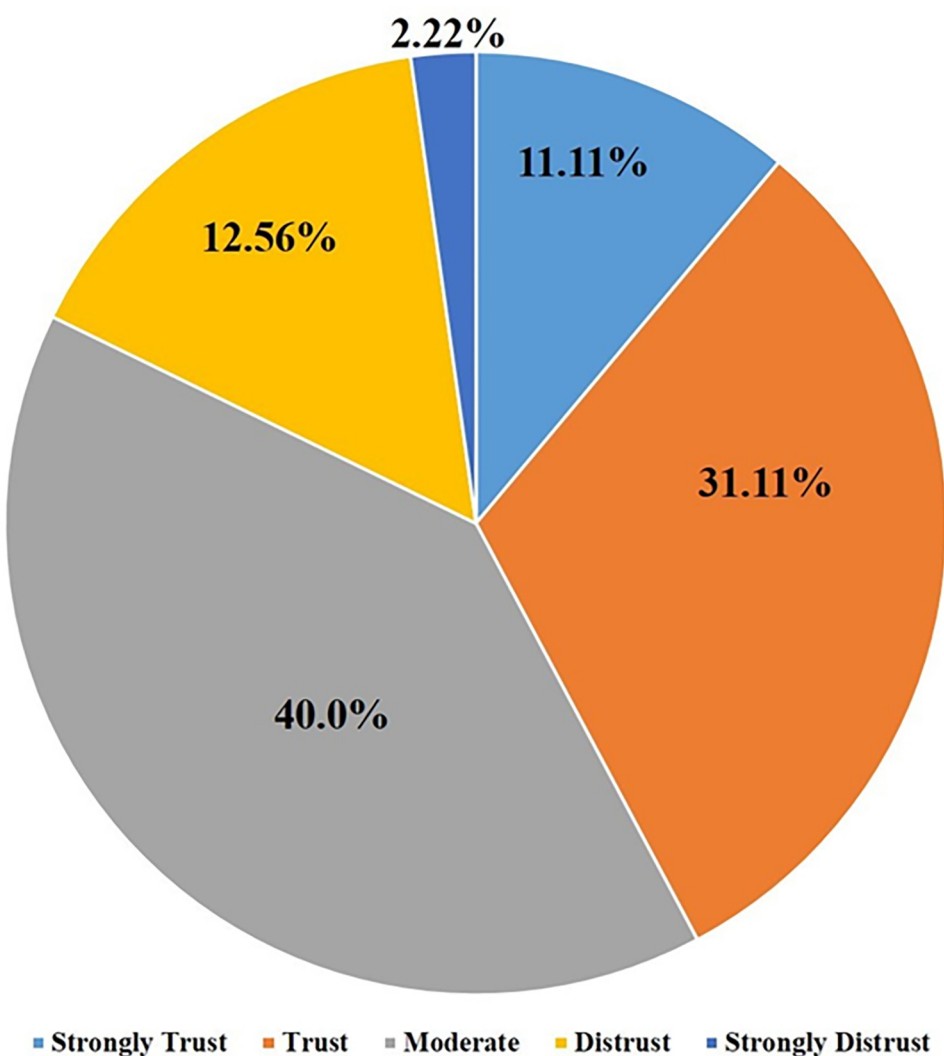

**Fig 3. Reliability of using O'view M to subjects.**

male infertility. Couples who do not go for testing must wait for the next and subsequent cycles, increasing psychological problems such as anxiety and depression.

Antonie van Leeuwenhoek in the 17th century was the first to visualize spermatozoa using single-ball handheld microscopes with high quality lenses that magnified objects 270-fold [19]. Using a smartphone camera with OVIEW-M magnified objects 230~250 fold. Smartphone cameras can magnify objects more than five-fold, with the camera and the OVIEW-M device magnifying objects 230~250 fold. This easy-to-use Smartphone-based semen analyzer could accurately measure sperm concentration, sperm motility, total sperm count, total motile sperm count, and changes in these parameters using small amounts of non-washed semen samples loaded onto disposable microchips. Use of these diagnostic assays can make male fertility tests as accessible, easy, fast, and private as pregnancy tests. Test results can be stored and imaged on smartphones and can enable tele-diagnosis [20].

This smartphone-based semen analyzer can also be used for home-based monitoring of men undergoing vasectomy and vasovasostomy. More than 33 million married couples worldwide prefer vasectomy for contraception, believing it to be safer, cheaper, and simpler than

female sterilization [21]. In the US, between 175,000 and 550,000 vasectomy procedures are performed per year [22]. Although the vasectomy failure is <1%, semen should be analyzed subsequently. Sperm concentrations in semen 8–16 weeks after surgery should be below 100,000/ml [23]. However, rates of postvasectomy follow-up semen analysis have been extremely poor [24]. The smartphone based semen analyzer may improve compliance with semen analysis after vasectomy by providing a fast, easy to use, inexpensive, and personalized method that accurately measures sperm concentration and motility. Similarly, the semen of men who have undergone vasovasostomy and varicocelectomy for infertility should be evaluated postoperatively to analyze sperm quality. The smartphone-based semen analyzer would also be applicable to these individuals.

Application to animals of breeding can be another way to using smartphone-based semen analyzer at home, laboratory, or zoo. Collected semen sample by animal breeders, they need to carry semen in certain condition with any harm, or trauma to check exact semen condition. However, using smartphone-based semen analyzer can give comfort to user without giving damage to semen. Animal breeder can make quick decision whether to go hospital or use the sample on research. Given the difference between human semen and animal semen in terms of sperm concentration, motility and sample volume, the current version of the smartphone-based semen analyzer should be extended for application in animals.

Considering that many men are only clinically tested for fertility problems at a relatively late stage, the use of such smartphone-based devices can improve patient care. In other medical fields such as obstetrics, pediatrics, hematology, or ophthalmology, smartphone technology is already being used to improve efficiency, patient satisfaction and empowerment with respect to smartphone [25].

There is a debate over the case where the quality of sperm is deteriorated by COVID-19 [26], and a tool to measure this is needed. In the era of COVID-19, any equipment which can easily measure for semen at home is needed, and for this, it must have an equal effect compared to existing measurements.

This study was conducted with a structure similar to that of RCT, but there is a limitation in that it was conducted without prior registration such as clinical trial. Although there are some limitations in this article, this tool can be use as the standard of the measurement about semen analysis in the era of COVID-19.

## Conclusion

This study showed that the smartphone-based semen test was as accurate and reliable as laboratory-based tests. These smartphone-based tests can be performed easily at home by men who visit the hospital for evaluation of infertility. This equipment can be used for measurement of semen in the era of COVID-19.

## Supporting information

**S1 File. Validity evaluation (accuracy evaluation).**
(DOCX)

## Author Contributions

**Conceptualization:** Ji Hoon Kim, Jung Ki Jo.

**Data curation:** Ji Hyoung Roh.

**Formal analysis:** Daegwan Kim, Hyang Mi Kim.

**Writing – original draft:** Kyu Shik Kim, Jung Ki Jo.

**Writing – review & editing:** Jung Ki Jo.

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
