## [Decision Letter · Decision Letter 0]

2 Jan 2022

PONE-D-21-36708Ability and Accuracy of the Smartphone-based O`VIEW-M® Sperm Test: Useful tool in the era of Covid-19.PLOS ONE

Dear Dr. Jung Ki Jo,

Thank you for submitting your manuscript to PLOS ONE. After careful consideration, we feel that it has merit but does not fully meet PLOS ONE’s publication criteria as it currently stands. Therefore, we invite you to submit a revised version of the manuscript that addresses the points raised during the review process.

We look forward to receiving your revised manuscript.

Kind regards,

Joël R Drevet, Ph.D.

Academic Editor

PLOS ONE

Journal Requirements:

 “Yes. This article was funded by the Bio & Medical Technology Development Program of the National Research Foundation (NRF) and funded by the Korean government (MSIT) [grant number 2019M3E5D1A01069353].”

“Ji Hoon Kim is the Chief Executive Officer of INTIN Corporation.”

Reviewers' comments:

Reviewer's Responses to Questions

**Comments to the Author**

1. Is the manuscript technically sound, and do the data support the conclusions?

Reviewer #1: Yes

Reviewer #2: Yes

2. Has the statistical analysis been performed appropriately and rigorously? 

Reviewer #1: Yes

Reviewer #2: Yes

3. Have the authors made all data underlying the findings in their manuscript fully available?

Reviewer #1: Yes

Reviewer #2: Yes

4. Is the manuscript presented in an intelligible fashion and written in standard English?

Reviewer #1: Yes

Reviewer #2: Yes

5. Review Comments to the Author

Reviewer #1: This is the multicenter study which is showing good correlation between the new smart-phone-based semen analyzing kit and manual semen analysis. The tool OVIEW-M seems to have good potential as a clinical tool and this article is showing interesting result. I have some suggestions and questions below.

- Why did the author use only manual semen analysis in hospital? Don’t they use CASA?

- Why is the abstinence 5 days? According to WHO guideline, it should be 2-7 days. If there is specific reason, the author should mention.

- From this result, readers can only know the mean and SD. The graph such as simple correlation etc. which is showing the actual plots should be added.

- This study cohort only includes healthy donors. It is questionable that OVIEW-M is also reliable as actual semen analysis in patient cohort. Author should mention about this issue in discussion.

- In page 9, there are already some home test kits available for semen test in the world (see example at: https://seem.life/en/), although they are not always so accurate but aimed to just notify the people to go to hospital. Author should discuss about those kits and its merit and demerit.

- According to the result, the tool is very accurate compared to manual semen analysis. Do the author think this can replace going to the hospital, or they just want to notify the result roughly and indicate to go to the hospital? Author should discuss more about future expectation for clinical use.

Reviewer #2: To evaluate the accuracy of smartphone semen analysis using O`VIEW-M, the authors investigated the relationship between smartphone semen analysis results and laboratory semen analysis of 200 men. Although we think this article is very interesting, there are fix points, comments and some of the questions in this paper.

1. Can O`VIEW-M devices and apps be used on all smartphones? Because smartphones use different optical cameras and software.

2. Do O`VIEW-M devices and apps automatically display semen test results? Or does the user need to count the sperm count on the screen?

3. Please indicate whether each user performed the inspection in a different place. This is because microscopes using a single ball lense are strongly influenced by the light source.

4. Was each test using O`VIEW-M done by an individual? Or did the representative perform?

6. PLOS authors have the option to publish the peer review history of their article (what does this mean?). If published, this will include your full peer review and any attached files.

Reviewer #1: No

Reviewer #2: No

---

## [Author Response · Author response to Decision Letter 0]

13 Apr 2022

Revision Letter

Dear Editor, Edrian Nim Tolentino

 We are pleased to submit to you a revised version of our manuscript #: PONE-D-21-36708, Title: "Ability and Accuracy of the Smartphone-based O`VIEW-M® Sperm Test: Useful tool in the era of Covid-19.” with hope of it being published in PLOS ONE. Followings are comments on our manuscript along with our replies.

Sincerely,

Corresponding author

The followings are our responses to the queries given by Reviewers.

1. We note that you provided the following response to our query regarding any patents or products in development related to this research: "Yes. The products in development or marketed products associated with this research are as follows. A measuring device of salive for smart phone (Reg. No. 10-1533343), The test device for Body fluid analysis using natural light (Reg. No. 10-1813866), CHAMBER FOR INSPECTING BODY-FLUID AND BODY-FLUID INSPECTING APPARATUS USING THE SAME (Reg. No. 10-0029588) and so on."

Please clarify what is meant by "and so on", and whether there are any additional patents, products in development or marketed products associated with this research to declare that were not included in your response above. 

“Yes. As you mentioned, we have confirmed that there are no additional patents, products in development or products on the market related to this research. Therefore, there are A measuring device of salive for smart phone (Reg. No. 10-1533343), The test device for Body fluid analysis using natural light (Reg. No. 10-1813866), CHAMBER FOR INSPECTING BODY-FLUID AND BODY-FLUID INSPECTING APPARATUS USING THE SAME (Reg. No. 10-0029588) products related to this research.”

2. Please review our data sharing policy: https://journals.plos.org/plosone/s/data-availability

PLOS journals require authors to make all data necessary to replicate their study’s findings publicly available without restriction at the time of publication. When specific legal or ethical restrictions prohibit public sharing of a data set, authors must indicate how others may obtain access to the data.

“Yes. We submit the raw data used in this study as a supplement to indicate how others may obtain access to the data.”

---

## [Decision Letter · Decision Letter 1]

30 May 2022

Ability and Accuracy of the Smartphone-based O`VIEW-M® Sperm Test: Useful tool in the era of Covid-19.

PONE-D-21-36708R1

Dear Jung Ki Jo,

We’re pleased to inform you that your manuscript has been judged scientifically suitable for publication and will be formally accepted for publication once it meets all outstanding technical requirements.

Kind regards,

Joël R Drevet, Ph.D.

Academic Editor

PLOS ONE

Additional Editor Comments (optional):

Reviewers' comments:

Reviewer's Responses to Questions

**Comments to the Author**

1. If the authors have adequately addressed your comments raised in a previous round of review and you feel that this manuscript is now acceptable for publication, you may indicate that here to bypass the “Comments to the Author” section, enter your conflict of interest statement in the “Confidential to Editor” section, and submit your "Accept" recommendation.

Reviewer #1: All comments have been addressed

Reviewer #2: All comments have been addressed

2. Is the manuscript technically sound, and do the data support the conclusions?

Reviewer #1: Yes

Reviewer #2: Yes

3. Has the statistical analysis been performed appropriately and rigorously? 

Reviewer #1: Yes

Reviewer #2: Yes

4. Have the authors made all data underlying the findings in their manuscript fully available?

Reviewer #1: Yes

Reviewer #2: Yes

5. Is the manuscript presented in an intelligible fashion and written in standard English?

Reviewer #1: Yes

Reviewer #2: Yes

6. Review Comments to the Author

Reviewer #1: All my comments included in my revision were now all addressed in the manuscript. I think this manuscript is ready for publication.

Reviewer #2: (No Response)

7. PLOS authors have the option to publish the peer review history of their article (what does this mean?). If published, this will include your full peer review and any attached files.

Reviewer #1: No

Reviewer #2: No

---

## [Editor Report · Acceptance letter]

6 Jun 2022

PONE-D-21-36708R1 

Ability and Accuracy of the Smartphone-based O`VIEW-M® Sperm Test: Useful tool in the era of Covid-19 

Dear Dr. Jo:

I'm pleased to inform you that your manuscript has been deemed suitable for publication in PLOS ONE. Congratulations! Your manuscript is now with our production department. 

Kind regards, 

on behalf of

Prof. Joël R Drevet 

Academic Editor

PLOS ONE